# Mapping the Concept, Content, and Outcome of Family-Based Outdoor Therapy for Children and Adolescents with Mental Health Problems: A Scoping Review

**DOI:** 10.3390/ijerph19105825

**Published:** 2022-05-10

**Authors:** Tonje H. Stea, Miek C. Jong, Liv Fegran, Ellen Sejersted, Mats Jong, Sophia L. H. Wahlgren, Carina R. Fernee

**Affiliations:** 1Department of Child and Adolescent Mental Health, Sørlandet Hospital HE, 4604 Kristiansand, Norway; sophia.hjorth@gmail.com (S.L.H.W.); carina.fernee@sshf.no (C.R.F.); 2Department of Health and Nursing Science, University of Agder, 4604 Kristiansand, Norway; liv.fegran@uia.no; 3National Research Center in Complementary and Alternative Medicine (NAFKAM), Department of Community Medicine, UiT, The Arctic University of Norway, 9037 Tromsø, Norway; miek.jong@uit.no; 4The Library, University of Agder, 4604 Kristiansand, Norway; ellen.sejersted@uia.no; 5Department of Health Sciences, Mid Sweden University, 851 70 Sundsvall, Sweden; mats.jong@miun.se; 6Department of Sport Science and Physical Education, University of Agder, 4604 Kristiansand, Norway

**Keywords:** scoping review, outdoor therapy, family-based therapy, adolescents, mental health problems

## Abstract

Outdoor therapy and family-based therapy are suggested to be promising interventions for the treatment of mental health problems. The aim of the present scoping review was to systematically map the concept, content, and outcome of combining family- and outdoor-based therapy for children and adolescents with mental health problems. The Joanna Briggs Institute methodology and PRISMA guidelines were applied. Eligible qualitative and quantitative studies were screened, included, and extracted for data. Seven studies were included. Findings from these studies indicated that family-based outdoor therapy programs have a positive impact on family- and peer relationships, adolescent behavior, mental health, self-perceptions (self-concept), school success, social engagement, and delinquency rates. However, participant characteristics, study design, and content and mode of delivery of the interventions varied substantially, hence preventing detailed comparison of outcomes across studies. In addition, most of the studies included few participants and lacked population diversity and comparable control groups. Although important ethical concerns were raised, such as non-voluntary participation in some of the programs, there was a lack of reporting on safety. This review indicates that a combination of family- and outdoor-based therapy may benefit mental health among children and adolescents, but due to the limited number of studies eligible for inclusion and high levels of heterogeneity, it was difficult to draw firm conclusions. Thus, future theory-based studies using robust designs are warranted.

## 1. Introduction

Prior to the COVID-19 pandemic, mental health conditions accounted for 16% of the global burden of disease and injury among youth, and half of these mental health conditions start by 14 years of age [1,2]. During and after the pandemic, however, children and adolescents have been more likely to experience higher rates of depression and anxiety [3]. The consequence of not addressing adolescent mental health conditions extends into adulthood, impairing both physical and mental health and limiting opportunities to lead fulfilling lives as adults [4]. Thus, it is critical that effective mental health strategies tailored to the needs of children and adolescents are explored.

Systematic reviews have reported findings that support the contention that nature positively influences mental health [5], and there is evidence supporting associations between exposure to nature and improved cognitive function, emotional well-being, brain activity, blood pressure, mental health, physical activity, and sleep [6,7,8,9]. Recontextualizing treatment from a hospital setting and into nature is considered a key component in successful therapeutic interventions for some youth and families [10]. Several reviews have reported beneficial health effects of nature-based programs [11,12]. Particularly beneficial capacities of the natural environment include mental and physical restoration through reducing psycho-physiological stress levels, increasing reflective capacities, as well as access to and regulating emotions [13]. In addition, shared outdoor activities involve physical mobilization along with numerous opportunities for social interaction, communication, and emotional reconnection between parents and their children [14].

A recently published umbrella review examining nature’s role in outdoor therapy has concluded that the evidence of treatment outcomes across outdoor therapies is mostly positive [8]. Furthermore, researchers have warranted the importance of examining how family involvement in adolescent outdoor treatment may strengthen the parent-child relationship and increase the self-efficacy of parents and caregivers [15].

Family-based treatment approaches are generally recognized to be highly effective for the treatment of several mental health conditions in children [16] and are also associated with a reduced prevalence of parental stress and depression [17]. There may be additional advantages to using a multi-family approach, compared to individual or single-family approaches, residing in the support, motivation, and connection generated both within and between participating families [18]. As such, multi-family groups have been described as a natural and useful format to promote change and growth [19]. Furthermore, it is hypothesized that a combination of using a family-based therapeutic approach in an outdoor environment could have a positive synergetic and sustainable effect on the mental health of children, adolescents, and their families [20,21,22]. However, there is a lack of reviews that have investigated the concept, content, and outcome of family-based outdoor therapy for children and adolescents with mental health challenges. Thus, the main aim of the present scoping review was to collect and systematically map information on family-based outdoor treatment targeting children and adolescents with mental health problems. Results from this study will inform researchers, practitioners, policymakers, and funders about experiences with family-based outdoor therapy programs.

## 2. Methods and Analyses

The present scoping review was conducted according to a predefined study protocol (not published). This scoping review used the methodological framework (five stages) developed by Arksey and O’Malley [23], which has been updated and advanced by Levac and co-workers [24] and further expanded by new guidance from the Joanna Briggs Institute (JBI) [25]. This framework provided the following recommendations for refining the methodology: (1) identification of research questions; (2) identification of relevant articles; (3) extraction of data/study selection; (4) charting the data; (5) collating, summarizing, and reporting results. All included studies have been described in a data charting form, and the scoping review adheres to the Preferred Reporting Items for Systematic Reviews and Meta-Analyses Extension for Scoping Reviews (PRISMA-ScR) [26]. A suitable team, with content and methodological expertise, was established early in the process to ensure the successful completion of the review. The first search for published or planned reviews on this topic was conducted in PubMed, Cochrane Library, JBI Evidence Synthesis, Epistemonikos, and Prospero. No previously published review focusing on similar target groups, methods, and outcomes were identified during this process.

### 2.1. Stage 1: Identification of Research Questions

As recommended by the JBI [25], research questions were derived from the Population-Concept-Context (PCC) mnemonic. For the present study, the *population* was children and adolescents aged 6–18 years old with self-reported mental health challenges or a mental health diagnosis and their families. At least one parent/caregiver had to be involved in the treatment. The *concept* was the underlying theoretical framework and content of family-based outdoor therapies, and the *context* was the study and intervention setting. Research questions were developed through discussions among members of the research team, which included therapists and researchers experienced in conducting and evaluating outdoor therapy targeting children and adolescents with mental health problems, and researchers with expertise in conducting scoping reviews. The following research questions were raised:

**Review question 1.** What are the characteristics of the primary child/adolescent participants (mental health status/diagnosis, age, socioeconomic status, referral agent) and their families that have participated in family-based outdoor therapy?

**Review question 2.** Which theoretical frameworks, therapeutic approaches, and outdoor activities are identified across the family-based outdoor interventions in the included studies?

**Review question 3.** In which ways and to what extent are family members included in the treatment?

**Review question 4.** Which professions (e.g., educational background, qualifications) are involved in outdoor therapy for children with mental health problems and their families?

**Review question 5.** What benefits and risks have been reported for children with mental health problems and their families participating in outdoor therapy?

**Review question 6.** Which barriers or facilitators for conducting family-based outdoor therapy targeting children with mental health problems have been identified?

**Review question 7.** What ethical issues or challenges related to the participation of children with mental health problems and their families in outdoor therapy have been identified?

### 2.2. Stage 2: Identification of Relevant Articles

#### 2.2.1. Search Strategy

In line with the JBI Manual for scoping reviews, a three-step search strategy was performed [25].

The initial search was executed in the following databases, MEDLINE, EMBASE, CINAHL, and APA PsycInfo. The results were then followed by an analysis of the text words (title, abstract, or authors keywords) of the retrieved articles, and of the index terms (subject heading words) used to describe the articles. An earlier review was also examined for words for the search concepts of wilderness-related therapies [27], followed by discussions and comments from all team members.

A second search using all identified text words and index terms for the three concepts, outdoor therapies, family or parents, and children or young people, was then undertaken by the librarian from the research team across all included databases, MEDLINE (Ovid), EMBASE (Ovid), APA PsycInfo (Ovid), AMED (Ovid), Scopus, CENTRAL trials database (Cochrane library), and CINAHL (EBSCOhost). The search was executed with no limitation regarding year or language. The final search strategy for all the databases is included in the Appendix A.

Thirdly, the reference list of the included studies was manually checked in the full-text screening process and double-checked by the team members who were extracting the data from the included studies.

#### 2.2.2. Eligibility Criteria

Eligibility criteria for inclusion or exclusion of studies were defined according to the broad Population-Concept-Context (PCC) mnemonic recommended by the Joanna Briggs Institute for scoping reviews [25]. In the present study, studies on family-based therapy taking place outdoors were included. This practice could be identified as adventure therapy, ecotherapy, outdoor rehabilitation, nature-based programs, wilderness programs, forest bathing, and/or bushcraft, as long as the role of nature was intended to have therapeutic benefits. Studies that focused on indoor therapy and had not evaluated outdoor experiences were excluded. Moreover, studies that were not offering a program involving both a primary child/adolescent participant with identified challenges associated with mental health problems and their family members, at a minimum one parent/caregiver, were excluded. Thus, studies focusing on the prevention of mental health problems, studies with no (primary or secondary) mental health outcomes, and studies that did not include a parent/caregiver were not included. Finally, peer-reviewed primary research studies of any study design reported in the English language were included, whereas reviews, commentaries, opinion pieces, conference proceedings, letters, editorials, trial registrations, evaluation reports, abstracts, theses, and book chapters were excluded.

### 2.3. Stage 3: Study Selection

The MEDLINE, APA PsycInfo, EMBASE, and AMED search strategies were run simultaneously in Ovid. The results were de-duplicated using the Ovid de-duplication tool before being exported to EndNote X9.3.3, together with the results from the other databases, and then the rest of the duplicates were removed (ES). Further, titles and abstracts were screened by three reviewers (THS, SLHW, MCJ) against the inclusion criteria using Rayyan software [28]. To assure quality in the screening process of titles and abstracts, all reviewers screened 100 randomly selected, identical studies based on the predefined inclusion and exclusion criteria. When inclusion or exclusion of a study could not be determined based on information in the title and abstract, a full-text screening of the reports was conducted. The result from the initial screening process was shared with three researchers (THS, CRF, MJ) who independently screened full-text articles for relevance according to the inclusion criteria. Results and disagreements between reviewers were discussed with co-authors (ES, LF, SLHW, THS, CRF, and MJ), and final decisions were made. Reasons for excluding studies were registered and documented throughout the process and reported in Figure 1.

### 2.4. Stage 4: Charting the Data

Data were extracted according to the PRISMA-ScR checklist [23,25], and a data charting form was developed to provide a descriptive summary of the results from all included articles. Corresponding to the predefined aims and review questions, study characteristics (author, year, country, design, and outcome measures), study aim and study population (sample and reason for referral), intervention description (context, concept, content, including professional involvement, and instruments used to measure outcomes), study findings (results, barriers, facilitators, and limitations) and ethical considerations were extracted in the extraction forms. Data were coded and entered in Microsoft Excel by the first author (THS) and thoroughly reviewed by two co-authors (LF, CRF).

### 2.5. Stage 5: Collating, Summarizing, and Reporting Results

Findings from the data charting form were discussed between two co-authors (THS and LF), and the first author (THS) was responsible for collating, summarizing, and reporting results, which were reviewed and revised by all co-authors. Findings were mostly described in a narrative style and presented in extraction tables and, where appropriate, in themes and text. Where numerical data of participant characteristics were available, descriptive statistics were used to summarize the data.

## 3. Results

### 3.1. Background Information about the Studies, Participants, and Outcome Measures

According to review question 1, the background characteristics of the primary child/adolescent participants and their families were identified. Background information about the studies included in the present scoping review, such as study aim, design, participant characteristics, and reported outcome measures, is provided in Table 1.

All included studies (*n* = 7) were from the United States and had an experimental design. Three of the studies included no control- or comparison groups, two studies included a comparison group, and one study included a control group. Participants were not randomly assigned to treatment/intervention groups and comparison- or control groups.

Primary child and youth participants ranged from six to eighteen years of age. Only two studies reported on the referral agent, which in both cases were the parents [29,30]. In one study [31], the use of coercion due to involuntary treatment was reported. The reasons for referral to treatment varied, but most participants were referred due to substance abuse, behavior problems/delinquent activity, poor family relationships, and/or emotional dysregulation/mental illness. As shown in Table 1, most of the studies included a higher number of males than females and predominantly families of Caucasian ethnicity. Those participating in the study by Norton and co-workers [32], who were primarily referred due to experiences of sexual abuse and a primary diagnosis of adjustment disorder, were, on the other hand, identified mostly as females of Hispanic and Caucasian ethnicity. Moreover, McLendon and Bandoroff [29] reported that they mainly included two-parent families, whereas the study of Pommier [2] reported that the included juvenile status offenders were predominantly living with either their mother or father. Only two studies included information about the socio-economic background of the participants. Bandoroff and Scherer [29] specified that mostly upper-middle-class families participated in their study, whereas families classified as low-income participated in the study by Norton et al. [32]. Bettmann and Tucker [31] expressed concern for low-income families regarding the financial costs associated with participation in private programs in the US context.

The study by DeMille and Montgomery [30] provided information about a single clinical case, but they did not describe the methodology used to collect and analyze data. All other studies presented in this review (*n* = 6) used a quantitative approach, in which most of them included tests or instruments that have shown to provide valid and reliable results (*n* = 5). However, the 60-item questionnaire used by Harper and co-workers [33] had not been psychometrically evaluated but was developed with practitioners for practical purposes. Two of the studies also conducted qualitative analyses to provide in-depth knowledge about participants’ subjective experiences and triangulate quantitative findings [32,33].

### 3.2. Program Characteristics

According to review questions 2, 3, and 4, underlying theoretical frameworks, therapeutic approaches, outdoor activities, family involvement, and professions represented among the program staff were identified. See Table 2 for details about program characteristics.

Six of the family-based outdoor programs described varied in length from three days to three months, whereas no information about the length of the program was provided by DeMille and Montgomery [30].

All studies included individual and/or group-based counselling processes. Therapy sessions and various challenging and skill-enhancing activities were included to reinforce behavioral change and improve the mental health and well-being of participants and their families. Bettmann and Tucker [31] and DeMille and Montgomery [30] also reported that participants in their studies could earn academic credits upon completing the program.

Some of the studies included a family component after finishing a wilderness program for struggling youth [29]. Other studies incorporated elements of family treatment throughout the program [30,31,34], either offering therapy tailored to parents and caregivers during the treatment period [2] or supporting and encouraging parents to undertake their own treatment process during the program period [33]. The study by Norton et al. [32] integrated a family-based approach into the program and did not offer separate activities for children and their parents.

The theoretical framework underlying the applied therapeutic approaches was only partially or poorly described in most of the studies (*n* = 6), especially in relation to the incorporation of nature and outdoor life in the treatment. The studies that included a separate program targeting adolescents, offered different challenging outdoor activities, such as hiking/backpacking and overnight camping, as well as learning outdoor/survival skills.

A presentation of professional involvement has also been presented in Table 2, which shows that none of the included studies specified the educational level and specialization of all staff involved in planning and implementing the programs.

#### 3.2.1. Findings


*According to review questions 5 and 6, benefits and risks for those participating in family-based outdoor therapy programs were identified, as well as barriers and/or facilitators to the implementation of the programs. See Table 3 for a description of key outcomes, reported barriers and facilitators, and study limitations described in the studies included.*


##### 
Key Outcomes


Whereas only a few studies (*n* = 3) reported negative outcomes (risks) associated with program participation, all studies reported some or several positive outcomes associated with participation in family-based outdoor therapy. Despite lack of statistical testing, results reported by Bandoroff and Scherer [29] indicated family functioning within the clinical range at pretest and within the normal range 6 months after completing the Family Wheel program and the standard wilderness program (comparison group). In addition, adolescent ratings of self-concept increased, adolescent ratings of delinquency dropped, and parent ratings of problem behavior improved during the intervention period in both groups. According to Bettmann and Tucker [31], adolescents with mental illness and/or substance misuse showed significantly improved attachment relationships in terms of decreased anger toward parents and increased emotional connection with both mother and father after participation in a wilderness program. At the same time, adolescents reported a decreased sense of caregivers’ availability, decreased empathy for caregivers’ feelings, decreased sense of security that parents understand their needs and desires, and decreased sense that parents are sensitive and responsive to their emotional states and concerns. Results from this study also suggested that the program was more successful for older than younger adolescents and more effective at improving parental attachment relationships among non-depressed adolescents and those without substance dependence, than adolescents with depression and those who were abusing or dependent on substances. Findings from the case narrative by DeMille and Montgomery [30] indicated that being outdoors had contributed to sustained improvement in family relationships. In addition, parents reported that the program helped them develop their parenting skills and to understand their son’s concerns, fears, and needs. Results from Harper and co-workers [33] indicated significant improvements in measures of family function, adolescent behavior and mental health, school success, and social engagement for both male and female participants 2 months after participation in a wilderness program, but improvement in school performance was more pronounced in males than females. However, despite significant improvements, results indicate that these issues persisted. Results from a 12-month follow-up showed a declined effect in some areas, whereas some items in the family function, mental health, and school performance construct remained improved compared to baseline results. According to McLendon and co-workers [34], preliminary results from their study combining family camp with regular treatment at a community mental health center indicated significant and clinically relevant improvement in family cohesion and family functioning. Moreover, all treatment group children made improvements according to the child behavior checklist, whereas comparison group children did not. The study by Norton and co-workers [32] reported that trauma-informed adventure therapy significantly reduced symptoms of anxiety and depression after participating in an adventure therapy program and receiving standard treatment at a trauma-focused care center compared to those only receiving standard treatment. Results from this study also indicated improved family functioning, especially in areas of communication, closeness, and problem-solving skills. Qualitative data supported these findings by providing information about the positive impact of the intervention on family communication, cohesion, and problem-solving, in addition to enhanced family interaction and competence building. The study by Pommier and Witt [2], targeting juvenile status offenders and their parents in an Outward Bound School program, showed a significant increase in family functioning, improvement in self-concept, and a reduction in behavioral problems in the intervention group compared to the control group at the end of the intervention period, four weeks after initiation of the program. However, a declining impact in several subscales reflecting all main outcomes was observed four months after initiating the program, and three months after the initial evaluation.

##### 
Identified Barriers and/or Facilitators


Bandoroff and Scherer [29] described that a barrier to completing the wilderness program was the high physical demands of desert living and that the intensity of the intervention entailed substantial risks and required commitment to the process by the participants. In addition, they reflect on the fact that the Family Wheel participants were often troubled families with high levels of marital discord and less capacity for cohesive family processes, which made the authors question whether these families may be less prepared for the challenges associated with wilderness therapy. The study by Bettmann and Tucker [31] indicated that some of the participants may have been involuntarily placed in treatment by their parents, which may have negatively affected participants’ susceptibility to treatment. Moreover, the weekly shift in staff and daily introduction of new adolescents, whereas other adolescents leave the program, may have negatively affected staff and peer attachment, according to the authors. The single case study by DeMille and Montgomery [30] reported that the participant initially refused treatment. To overcome this barrier to treatment, the authors described that they used the first sessions to focus on developing a working relationship, help the participant feel safe, and develop hope of improving relationship quality and quality of life. Due to an adverse reaction to the use of diagnostic labels, treatment planning included a functional approach, and the participant reported that over time he was able to identify negative aspects of life that he wanted to change. From his parents’ perspective, the use of a narrative therapeutic approach was the most helpful aspect of the treatment process. McLendon and co-workers [34] reported difficulties to recruit comparison families receiving regular treatment at a community mental health center and collect complete data sets from this group. On the other hand, the authors described that family-directed structural therapy was well suited for family camp because it gathered much information and provided a structure to camp therapy sessions. No other barriers or facilitating factors associated with conducting family-based outdoor therapy programs were identified in the included studies.

Additionally, study limitations reported by the authors of the included studies are presented in Table 3.

#### 3.2.2. Ethics


*According to the final review question 7, ethical considerations were identified that related to the participation of children with mental health problems and their families in the included outdoor programs.*


Most of the studies (*n* = 4) did not report ethical considerations in their papers. Only Norton and co-workers [32] reported that their study was approved by the appropriate research ethics committee and was performed in accordance with the ethical standards described in the Declaration of Helsinki.

An ethical consideration mentioned in the study by Bandoroff and Scherer [29] was that some of the participants may have felt betrayed when the positive achievements families had experienced in the wilderness were not sustained after returning to everyday life at home. Bettman and Tucker [31] also mention the potential doubt on the viability of wilderness therapy programs due to previously reported detrimental incidents, including neglect and in a few cases also fatalities, and that some participants in the included study may have been involuntary transported to treatment. Moreover, the high treatment costs of wilderness programs were also questioned and expected to explain the low participation rates of low-income families. Finally, Pommier and Witt [2] argued that although it would have been ideal to include an additional group participating in the OBS program, but not the family intervention program, the leaders were, for ethical reasons, not willing to offer less than the full amount of available intervention.

## 4. Discussion

The present scoping review is the first to broadly map the concept, content, and outcome of studies presenting family-based outdoor therapy for children and adolescents with mental health problems. This review identified seven studies targeting children and adolescents experiencing mental, emotional, developmental, behavioral, or social difficulties, and their families using different types of nature-based therapy interventions. The majority of studies (*n* = 4) were based on wilderness therapy programs, but all programs included in this review varied substantially in lengths of stay, amount of time spent in nature, types of outdoor environments, involvement of qualified therapists, therapeutic approaches, and degree of family involvement. In addition, the studies had different types of quasi-experimental repeated measure designs, and used a quantitative, qualitative, or mixed methods approach for data collection and analyses. Findings reported from these studies indicated that family-based outdoor therapy programs have a positive impact on a range of different outcomes, including family- and peer relationships, adolescent behavior, mental health, self-perceptions (self-concept), school success, social engagement, and delinquency rates. However, differences in participant characteristics, study design, and content and mode of delivery of the programs varied substantially, hence preventing a detailed comparison of outcomes across studies. Moreover, the question of whether the positive outcomes are due to the family involvement, the outdoor therapy setting, or the combination of these factors cannot be answered due to limitations in study designs.

Despite differences in participant- and program characteristics, an overall improvement in family relationships, measured as family functioning, family cohesion, and attachment relationship were observed in all studies after the program period. These findings were expected, as family involvement has been identified as an additional value to adolescent therapy programs [35], and structured outdoor family recreation programming has shown a positive effect on family strength [36]. A comprehensive review has also provided evidence supporting the effectiveness of systemic interventions, which include family therapy or other family-based approaches, for recovery from child abuse and neglect; conduct problems, emotional problems, eating disorders, somatic problems, and first episode psychosis [16]. Findings reported in this review, which may be partly explained by the involvement of parents/caregivers in the process, are strongly supported by family systems theory which predicts and explains how people within a family system interact, and how interactions inside the family system are different from those outside of it [37].

An unexpected outcome presented in this review, however, was that adolescents participating in the study of Bettman and Tucker [31] also reported negative relationship outcomes reflecting less trust and communication with parents and peers by the end of treatment. According to the authors, the mixed results shown in their study may be partly explained by involuntary treatment admission and feeling of parental rejection, where out of home placement may have negatively impacted attachment relationships. Results from another study, however, have indicated that the use of forcible transport did not affect program outcomes among youth participating in residential care programs [38], but the validity of these results has been questioned [39].

### 4.1. Ethical Concerns

Voluntary participation is linked to increased intrinsic motivation to change, whereas involuntary treatment, use of coercion, and transport services specialized for “uncooperative youth” in outdoor behavioral healthcare (wilderness therapy) have raised ethical and empirical concerns [39]. Despite ethical concerns, adolescents are still largely admitted involuntary to outdoor behavioral healthcare in the United States [40]. Bettman and Tucker [31] also mentioned the potential doubt about the viability of wilderness therapy programs due to previous reports of neglect, abuse, and fatal incidents. Increased focus, however, has recently been directed towards pursuing an ethic of care for outdoor therapy based on the human rights of the participants that furthermore emphasizes relational dignity, not only between humans but also in the human-nature relationship [41]. Adolescents’ self-determination and active choice to participate are suggested to function as a catalyst for change in wilderness therapy—or *friluftsterapi*—in a Norwegian context [42]. Another ethical concern reported by Bandoroff and Scherer [29] was related to possible feelings of betrayal of participating families due to a lack of sustainable positive impact after the program period. In line with these findings, other researchers have highlighted the importance of providing appropriate aftercare upon returning home [43]. Finally, the high treatment costs of wilderness programs have been recognized as a prohibitive factor for the participation of low-income families, which may also partly explain the lack of population diversity in terms of ethnicity and socioeconomic status.

Although important ethical concerns were raised in the studies included in the present review, there is a lack of reporting on safety, and few adverse effects associated with program participation were reported. Similarly, a systematic review has confirmed that outdoor therapy studies have a tendency to report only ‘good interactions’, and highlighted the need to identify potential risks and negative experiences associated with participation [44].

### 4.2. Gaps in Literature

Although results presented in this review indicate that family-based outdoor therapy represents a promising avenue for the treatment of mental health problems among children and adolescents, there are significant literature gaps that should be mentioned.

Although some of the studies presented in this review refer to theoretical frameworks, it is not clear how these frameworks have been operationalized. To a certain extent, the authors have presented theoretical frameworks supporting that parental involvement is an important factor in outdoor therapy targeting children and adolescents with mental health problems. However, in line with the results presented in a recent umbrella review, few previous studies have described how theoretical frameworks have contributed to identifying underlying mechanisms explaining the role of nature in outdoor therapy programs [12].

Further, a systematic review focusing on outdoor interventions for health and wellbeing suggested that future studies should use a more inter-disciplinary framework such as a complex systems approach that considers the complexity of multiple stakeholder groups and how they simultaneously affect and are affected by the multi-dimensional nature of outdoor therapy [44].

The studies described in this review had identified important barriers, but information about facilitating factors associated with program implementation and participation was scarce. As they are dealing with vulnerable groups with complex health challenges and behaviors, an increased understanding of the factors that facilitate and hinder intervention sustainability is needed to inform future research and clinical practice.

However, low participation rate, differences in participant characteristics, a broad spectrum of program characteristics, and several methodological limitations characterizing the studies presented in this review prevent detailed recommendations for clinical practice. A broad spectrum of program characteristics has previously been described within the field of adventure therapy [11], and another scoping review on nature-based interventions for vulnerable youth has confirmed the lack of studies using a methodologically robust empirical design to increase external validity [45]. Thus, future studies targeting more specific groups, including true comparison groups, and using methodological sound instruments for data collection and analyses are warranted. However, it is important to be aware that although utilization of randomized controlled designs has been recommended when possible and appropriate, randomization is not always feasible, the inclusion of control groups is not always ethically acceptable [46], and the outdoor environment is less controllable than traditional indoor treatment environments [2,12,44].

Due to the limited cultural and socioeconomic diversity of those participating in the presented studies, and lack of long-term follow-up, it is also unclear whether family-based outdoor programs would have a different impact on different groups and have a sustainable impact after the program period.

Another limitation reported in this review, which limits the possibility to make conclusions regarding program impact, is that all included studies used self-report measures, and some did not use validated questionnaires or report methods used for data collection and analyses. Thus, it is evident that there is a need for future studies to develop and validate instruments that provide important information about relevant outcomes from such studies, and especially instruments that provide information about the impact of nature contact during therapy. Moreover, the use of objective measurements of vital body functions reflecting stress level and immune function has been warranted as health response indicators of nature-based therapeutic intervention studies [47].

### 4.3. Evidence for Practice

Due to the limited number of studies presenting family-based outdoor therapy programs for vulnerable children and adolescents, the variation in characteristics of participants, and methodological limitations of the presented studies, the evidence base for practice is scarce. Thus, clinicians need to acknowledge the very limited evidence currently available to support the practice of family-based outdoor therapy and rely more on research with other populations and concepts until the field is further advanced. Previous reviews have reported beneficial effects of nature-based programs [11,12], and family-based therapy is recognized to be an effective treatment approach for mental health conditions in children [16]. Although most of the studies included in this scoping review are associated with positive outcomes, more high-quality studies are warranted in order to increase the knowledge base of the outcomes associated with this innovative approach to improving mental health and wellbeing among struggling children and adolescents.

It is also important to report whether different programs are more successful for certain sub-groups. Among the studies included in the present review, Bandoroff and Scherer [29] showed that wilderness family therapies seemed to be more beneficial for younger adolescents with a less severe history of behavioral disturbance. Based on experiences with a high number of participating families with a high level of marital discord, Bandoroff and Scherer [29] further argued that dysfunctional families are less suited for intensive wilderness treatment due to the lack of capacity for cohesive family processes. Previous findings have indicated that parenting stress and couples’ relationship quality are empirically related [48]. Furthermore, participant characteristics presented in the study of juvenile status offenders by Pommier and Witt [2], also indicate a relationship between parental marital status and child difficulties. Thus, information about parental relationships and the level of family dysfunction would most likely be highly relevant in order to provide tailored therapy sessions according to each family’s needs.

### 4.4. Strengths and Limitations

A strength of this present scoping review is that a comprehensive search was conducted in key databases, transparently documented. Reference lists were also subsequently reviewed. However, it holds some limitations as a grey literature search was not conducted and the inclusion of peer-reviewed studies was limited to the English language.

As scoping reviews do not require critical appraisals, no quality assessment was performed on the included studies. However, from the overview of study limitations in Table 3, several methodological shortcomings were identified by the authors themselves, that need to be taken into account in future studies. The low participation rate, lack of cultural and socio-economic diversity among participants, differences in mental health challenges among participants, differences in therapeutic approach and activities, and different therapeutic environments prevented a detailed comparison of outcomes between the studies. Moreover, the generalizability of results presented in this scoping review to another context is questionable, as all the included studies were conducted on programs in the United States. The lack of studies from other nations than the United States has been identified as a general challenge within the existing literature on outdoor adventure practices or research among children and adolescents [49]. Finally, two of the included studies date back to the early 1990s. Although representing pioneer studies in terms of integrating systemic family approaches and wilderness therapy, the use of language when referring to adolescent participants as “problem youth” and equivalent labels is outdated and unacceptable. According to Harper and Fernee [41], outdoor therapy in the 21st century ought to be sensitive to the particularity of situations and the subtle ways in which people may be excluded, marginalized, disrespected, or devalued. A resource-focused approach abstains from or minimizes, using predefined categories, pathologies and labels. Furthermore, an ecological and systemic framework is concerned with the complex webs of relations in which a child, adolescent, or parent is situated and remains attuned to the participants’ emotional attachments, particular needs, and vulnerabilities in order to maintain relational dignity in family-based outdoor therapy.

## 5. Conclusions

This is the first scoping review that has mapped the development and implementation of family-based outdoor therapy among children and adolescents with mental health problems. Only seven studies met the criteria to be included in this review, and the programs described were characterized by a diverse study population, setting, and activity. Most of the studies had a quantitative design and all studies were conducted in the United States. Outcomes were largely positive across a wide range of psychosocial and behavioral measures and often maintained post-treatment. Despite a limited knowledge base, the findings presented in this study provide important insight into the benefits and risks of family-based outdoor therapy programs targeting children and adolescents with mental health problems. This scoping review will inform health care professionals and researchers and guide the development and implementation of family-based outdoor therapy. Future studies should be tailored to reach different sub-groups, use robust empirical designs, provide a comprehensive theoretical and practical description of the intervention, and use validated methods for data collection and analyses in order to build the evidence base for systemic outdoor therapy. Finally, there is a need for studies originating in countries outside the United States to increase the knowledge and experience of family-based outdoor therapy in other contexts and regions of the world.

## Figures and Tables

**Figure 1 ijerph-19-05825-f001:**
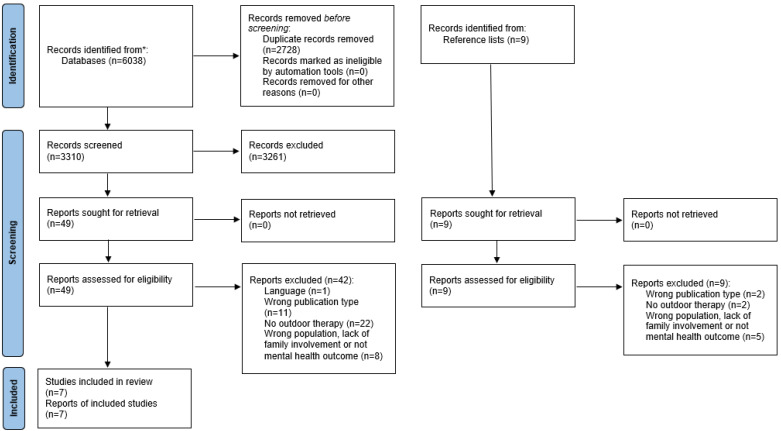
PRISMA flow diagram, from [26]. * Records identified from Ovid MEDLINE (*n* = 921), Ovid EMBASE (*n* = 1483), Ovid APA PsycInfo (*n* = 1080), Ovid AMED (*n* = 31), CINAHL (*n* = 676), CENTRAL (Cochrane library) (*n* = 308), Scopus (*n* = 1539).

**Table 1 ijerph-19-05825-t001:** Background information about study aim, study design, participants, reason for referral, and outcome measures.

Author, Year, and Country	Study Aim	Study Design	Study Sample	Reason for Referral	Outcome Measures/Instruments *
Bandoroff & Scherer, 1994, United States	To enhance perceptions of family functioning, reduce problem behavior, and improve self-concept among adolescents through participation in wilderness family therapy following standard wilderness therapy, compared to those only receiving wilderness therapy.	Experimental design with a comparison group (treatment as usual). Non-random assignment. Pretest, 21-day post-test, and 6 weeks follow-up tests post-treatment.	Intervention group: 27 families with adolescents, 13–18 years old. Comparison group: 39 families with adolescents, 13–18 years old. 65% were males in the total sample.	Primarily referred due to substance abuse, behavior problems, poor school performance, and delinquent activity.	**Family functioning**: FAM III (adolescent and parents) **Self-concept**: SDQ III (adolescent) **Problem behavior**: SRDC (adolescent) RBPC (parent)
Bettmann & Tucker, 2011, United States	To examine shifts in adolescents’ attachment relationships with parents and peers during a wilderness therapy program.	Experimental one-group design. Pretest and 7-week post-test.	96 adolescents, 14–17 years old, and their families. 61.5% were males in the total sample.	Primarily referred due to Oppositional Defiant Disorder, Depressive Disorder, ADHD, and/or Substance Dependence/Abuse.	**Adolescent attachment:** AAQ (adolescents) **Adolescent unresolved attachment:** AUAQ (adolescents) **Parent and peer attachment:** IPPA (adolescents)
DeMille & Montgomery, 2016, United States	To illustrate the application of Narrative Family Therapy techniques in an Outdoor Behavioral Healthcare program.	Single case experimental study.	A 16-year-old male and his parents.	Referred due to emotional dysregulation, poor family relationships, and academic problems.	**An exit interview** (adolescent) was conducted, and **a questionnaire** was used to collect data post-intervention (adolescent and parents). No information about included instruments was provided.
Harper et al., 2007, United States	To examine changes in family functioning, adolescent behavior, and mental health issues following participation in a wilderness therapy program.	Experimental one group design. Pretests, 2- and 12-months follow-up tests post-treatment.	221 adolescents, 13–18 years old, and 124 parents. 62% were males in the total sample.	Referred due to emotional, behavioral, and substance use problems.	The following constructs were assessed in a 60-item questionnaire that had not been psychometrically tested: - Family function - Adolescent behavior - Adolescent mental health - School success - Social engagement Qualitative data were collected via **observation and focus groups**.
McLendon et al., 2009, United States	To determine the impact of a family-based program including a therapeutic wilderness camp in addition to usual counselling, compared to families receiving only usual counselling from a community mental health center.	Experimental design with a comparison group (treatment as usual). Non-random assignment. Pretest (at camp), 6-weeks and 6-months follow-up tests post-treatment.	Intervention group: 25 families; 52 children, 6–17 years old, and 41 adults. Comparison group: 15 families; 31 children, 8–20 years old, and 26 adults. No information about gender distribution.	Primarily referred due to behavior problems among seriously emotionally disturbed (SED) children or a problematic adult relationship. Comparison group families included at least one child diagnosed with SED.	**Family functioning**: FACES II(parents and children) **Competence and problems in children**: CBCL (parents)
Norton et al., 2019, United States	To determine the impact on child trauma symptoms and family functioning in a family-based program combining adventure therapy and usual counselling service.	Experimental design with a comparison group (treatment as usual). Non-random assignment. Pre- and 3-months post-tests were supplemented by qualitative data.	Intervention group: 18 children, 8–17 years old, and their families. Comparison group: 14 children, 8–17 years old, and their families. No information about gender distribution.	Primarily referred due to experiences of sexual abuse and a primary diagnosis of adjustment disorder.	**Impact of trauma**: TSCC (children) **Family functioning**: FAD (caregiver) Qualitative data were collected via **focus groups**.
Pommier & Witt, 1995, United States	To determine the impact on self-perception, behavior, and family functioning of an Outward Bound School program for adolescents that included a family training component.	Experimental with a control group. Non-random assignment. Pre-test, 4-weeks post-test, and 4-months follow-up test after the start of the treatment program.	Intervention group: 39 adolescents, 13–17 years old, and their families. Control group: 40 adolescents, 13–17 years old, and their families. 55.7% were males in the total sample.	Juvenile status offenders.	**Self-perception**: SPPA (adolescents) **Family functioning**: FACES II **Behavioral problems**: ECBI (parents)

***** FAM III—The family Assessment Measure III, SDQ III—The Self-Description Questionnaire III, SRDC—The Self-reported Delinquency Checklist, RBPC—The Revised Behavior Problem Checklist, AAG—The Adolescent Attachment Questionnaire, AUAQ—The Adolescent Unresolved Attachment Questionnaire, IPPA—The Inventory of Parent and Peer Attachment, FACES II—The Family Adaptability and Cohesion Evaluation Scale II, CBCL—The Child Behavior Checklist, TSCC—The Trauma Symptom Checklist for Children, FAD—The Family Assessment Device, SPPA—The Self-Perception Profile for Adolescents, ECBI—The Eyberg Child Behavior Inventory.

**Table 2 ijerph-19-05825-t002:** Family-based Outdoor Program characteristics.

Author	Program Structure	Program Framework, Approach, and Activities for Adolescents	Family Involvement	Professional Involvement
Bandoroff & Scherer, 1994	(1) **A 21-day adolescent wilderness program**. The main focus of this paper was part 2 of the program. (2) **A four-day family program**. Three main structural components: (1) a theme representing a critical family resource, (2) individual therapy sessions with families, and (3) multi-family therapy.	** Framework/approach ** : - Family systems theory - Family resource-focused approach following a wilderness program. ** Activities adolescents ** : - High desert terrain expedition - Daily hiking over several miles - Learning primitive living skills - Final 3 days were spent alone.	- Multi-family trekking - Family therapy sessions - Multi-family group discussions - Metaphorical exercises	(1) **Therapist**: visited the group on three occasions and conducted individual sessions with each student. (2) **Two therapists** (one was present for all trials). **Additional therapists** (advanced graduate students in clinical psychology with theoretical and practical training in structural family therapy). **Licensed clinical psychologist** and **structural family therapist**: served weekly as a supervisor and twice as a therapist.
Bettmann & Tucker, 2011	(1) **A seven-week adolescent wilderness program** (*n* = 9). Individual treatment plans, individual and group psychotherapy 2 days/week in addition to psychiatric consultations as needed during the program. Elements of family treatment were incorporated throughout the program. Academic credits were earned upon completion of the program. (2) **A three-day family program**. Including a 1-day psycho-educational parenting workshop and a 2-day therapeutic wilderness experience with youth.	** Framework/approach ** : - Not specified ** Activities adolescents ** : - Hiking to primitive campsites - Daily living tasks - Daily participation in an experimentally based academic curriculum	- Weekly family therapy at home - Regular phone contact between the family’s home therapist and the adolescent’s program therapist - Weekly therapeutic assignments and letters written and sent by adolescents to parents and vice versa - A 3-day family therapy process at the wilderness site at the end of the program period: a 1-day psycho-educational parenting workshop and a 2-day therapeutic experience in the wilderness with the child.	**Master-level clinician**: initial clinical screening **MD or PA**: initial medical screening **Master-/Doctoral-level clinicians**: creating treatment plans and providing individual and group therapy to adolescents twice/week. **Additional role for clinicians**: psychiatric consultation when needed during the program and aftercare planning together with each family.
DeMille & Montgomery, 2016	(1) **Adolescent wilderness program. Length not specified**. Only general features of the Outdoor Behavioral Healthcare program were described and no specific information about components was used in this case study. In general, the OBH program included individual, group, and family therapy combined with wilderness living within small peer groups. Academic credits were earned upon completion. (2) **End of trails family ceremony**.	** Framework/approach ** : - Narrative framework - Narrative family therapy technique in an OBH setting ** Activities adolescents ** : - Hiking and backpacking, on average four-five times/week for three-five miles each trek - Setting up/breaking down campsites - Practical wilderness skills relevant to their living situation, such as primitive fire making for warmth and preparing meals	- Both in-person and at a distance - Weekly meetings with the therapist via conference call - A series of group and family therapy sessions with and without their child - A series of shared narrative writing assignments - An “end of trails” ceremony, where parents visit and go camping with their child.	No details about the personnel responsible for treatment were provided.
Harper et al., 2007	**A 21-day wilderness therapy program** for the treatment of emotional, behavioral, or substance use diagnoses. In general, these programs include groups of seven youth and a treatment team.	** Framework/approach ** : - Family systems theory - Wilderness therapy guided by family systems approach ** Activities adolescents ** : - Challenging activities (e.g., expedition backpacking, rafting) - Intensive outdoor living conditions (e.g., avoiding hypothermia, and dehydration) - Active participation in individual and group counselling processes	- Staff encourages adolescents and family members to work with therapists to identify issues and treatment goals - Pre- and post-treatment meetings with therapists and field staff - Parents supported to undertake their own treatment process during the program period - Families and clinical team collaborate on discharge, transition, and aftercare planning.	** Expedition treatment team: ** **Clinical supervisor** **Medical supervisor** **Field therapists** **Wilderness leaders**
McLendon et al., 2009	**Three-day family camp** (*n* = 3–5). The family camps were provided **as an adjunctive to regular treatment** at a Community Mental Health Center. A half-day follow-up meeting between staff and families six weeks after each camp to discuss progress, provide positive reinforcement, and address possible struggles.	** Framework/approach ** : - Structural family therapy framework - Combines strength-based, group work and structural approaches in a wilderness setting and regular treatment ** Activities adolescents ** : - 3 three-hour child psychosocial groups during camp - Children attended psychosocial groups at follow-up	- Parents attended 3 three-hour family-directed structural therapy (FDST) groups during camp, while child groups were held concurrently. - Parents attended FDST groups at follow-up. - Three group sessions were conducted involving all family members. One of these family groups included an adventure-based activity.	**One lead therapist** **Two-three adjunct therapists** **Two-four child case managers**
Norton et al., 2019	**A three-month adventure therapy program** with the whole family in a community-based setting **as an adjunct to regular treatment** at the trauma-focused care center (ChildSafe).	** Framework/approach ** : - Trauma-informed framework - Multi-family trauma-informed adventure therapy	- “Talk therapy” in individual, group, and family settings - Hiking and camping outdoors - Kayaking, geocaching, archery, hiking, low and high ropes courses, rock climbing, and camping	**Mental health clinicians**: Trained in Trauma-Focused Cognitive Behavioral Therapy (TF-CBT).
Pommier & Witt, 1995	**A 30-day Outward Bound School program** for juvenile status offenders (*n* = 8–12) which included a family component. Six program stages: intake period (14 days), orientation period (2 days), expedition period (16 days), reunion period (10 days), reinforcement period (10 days), and facilitation period (14 days).Individual contracts with program goals to ensure lasting positive effects after the program period.	** Framework/approach ** : - Outward Bound framework - A multi-modal approach includes parental training/therapy, youth expedition, and follow-up. ** Activities adolescents ** : - River-based activities in a natural environment area (e.g., canoeing) - Activities to increase survival skills outdoors (e.g., camping, first aid, nutrition, and nature appreciation) - Individual conferences and contract development - Games and activities to reinforce behavioral changes	** Family involvement ** : - *Orientation*: parental education seminars. Parents and students set goals for behavior change - *Expedition*: workshops for parents - *Reunion*: individual follow-up and contract for sustained effects at home - *Reinforcement*: 3 family visits and 7 phone visits. A contract specifying changes that need to occur in the home, school, and community - *Facilitation*: Follow-up phone calls, letters, home visits (if needed)	**An instructor** (with a minimum of 140 days of relevant work experience) **An assistant instructor** **An intern** **A course director**: supervising each team in the field

**Table 3 ijerph-19-05825-t003:** Key outcomes, reported barriers, facilitators, and study limitations.

Study	Positive and Negative Outcomes (Benefits and Risks)	Identified Barriers and/or Facilitators	Reported Study Limitations (by Authors)
Bandoroff & Scherer, 1994	**Positive outcomes** (Treatment- and comparison group) - Improved family functioning - Reduced delinquency rates - Reduced problem behavior - Increased ratings of self-concept **Negative outcomes** - None reported	**Barriers** - Demanding physical program conditions - High-intensity program - Troubled families with high levels of marital discord **Facilitators** - None reported	- Self-selection of participants - Low participation rate - Small sample size - Lack of controls, randomization, and long-term follow up - Lack of cultural and socioeconomic diversity
Bettmann & Tucker, 2011	**Positive outcomes** - Improved attachment relationship - Increased emotional connection **Negative outcomes** - Decreased sense of caregivers’ availability - Decreased empathy for caregivers’ feelings - Decreased sense of security that parents understand their needs and desires - Decreased sense that parents are sensitive and responsive to emotional states and assist with concerns	**Barriers** - Involuntary treatment - Out of home placement may have negatively affected the attachment relationship - Weekly shift in staff - Daily replacement of program participants **Facilitators** - None reported	- No information about non-respondents - Lack of long-term follow-up - Lack of cultural and socioeconomic diversity
DeMille & Montgomery, 2016	**Positive outcomes** - Sustained improvement in family relationship - Helped parents understand their sons’ concerns, fears, and needs - Helped parents develop their parenting skills **Negative outcomes** - None reported	**Barriers** - Adolescent in denial of their own treatment needs **Facilitators** - Focused on developing a working relationship, helping him feel safe and develop hope of improvement - Use of a functional approach in treatment planning due to an adverse reaction to the use of diagnostic labels - Use of narrative as a learning tool - Time outdoors to identify the need for change	- A single case description - Methodology for collection and analyses of data was not reported
Harper et al., 2007	**Positive outcomes** - Improved family functioning - Improved behavior - Improved mental health - Improved school success - Improved social engagement **Negative outcomes** - Declined scores at 12 months follow-up	**Barriers** None reported **Facilitators** None reported	- Non-utilization of control groups for randomization of treatment - Instrument used for data collection has not been psychometrically tested
McLendon et al., 2009	**Positive outcomes** (Treatment- versus comparison group) - Improvement in family cohesion - Improved family functioning - Improved behavior among the treatment group, but not in the comparison group **Negative outcomes** - None reported	**Barriers** - Reported difficulties to recruit comparison families and collect complete data sets from them. **Facilitators** - Family-Directed Structural Therapy was well suited for family camp because it gathered much information and provided a structure to camp therapy sessions	- Small sample size - Non-randomly selected participants - Lack of cultural and socioeconomic diversity - The community mental health center system did not allow for the collection of child behavior data at exact time points according to the study protocol
Norton et al., 2019	**Positive outcomes** (Treatment- versus comparison group) - Reduction in trauma symptoms - Improved family functioning Supportive qualitative data: - Positive impact on family communication, cohesion, and problem-solving in addition to enhanced family behavioral and skill-building **Risks** - None reported	**Barriers** None reported **Facilitators** None reported	- Small sample size - Non-random selection of participants - No comparison group receiving no services - No long-term follow-up past three months
Pommier & Witt, 1995	**Positive outcomes** (Treatment- versus control group) - Improved self-perceptions (self-concept) - Improved family functioning - Improved behavior **Negative outcomes** - Declining scores at three months post-program	**Barriers** None reported **Facilitators** None reported	- Small sample size - Not comparable control group - Short follow-up period - Non-random convenience sample - Results reflecting behavioral change were only available for the treatment group - No group receiving only the OBS program

## Data Availability

Not applicable.

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
