# Peer review of "Mapping the Concept, Content, and Outcome of Family-Based Outdoor Therapy for Children and Adolescents with Mental Health Problems: A Scoping Review"

_ijerph, 2022, doi:10.3390/ijerph19105825_

Round 1

Reviewer 1 Report

This paper is interesting and well explain and reviews the literature on family-based outdoor therapy.

I would recommend this paper for publication but I do have several issues to address before submission.

The main thing is that I am wondering whether the authors can provide a meta-analytic approach (forest plots) on the quantitative studies included to have a clearer idea of the strength and overall effects of the therapy while controlling or contrasting age group, long vs. short term therapy, etc...

A second point is that only 7 studies have been included revealing a major gap in the literature and the possible lack a representativity of the findings. The authors highlighted this point in the discussion and conclusions but I think it should be added to the abstract.

Minor comments:

Several typos issues: L23 (space); Line 24 to 27 (font size); L62; L66; L90; L166

L51: There are much more articles on the topic including several meta-analysis on the positive impacts of natural sounds or natural landscapes. I think they should be cite here.

L105: What do you mean by "no protocol for a similar systematic review was found"? I think this is pretty standard. Are you talking about the methodology or about the research questions? Please be more specific.

L110: Why excluding physical health? (Ex: vitality, mobility etc...)?

L168: Can you briefly describe what is PCC mnemonic?

Figure 1: From what I understand, the paper added from references list have been added after screening. They should be added as other sources at the identification stage. Category with N=0 should be removed. In the inclusion part, the N=7+ N=7 is misleading, please choose only one category.

L227: If possible move this part to supplemental material.

L267: check grammar?

L275: as mentioned before, might be a nice add to perform a meta-analysis of quantitative studies (pre-post SMD). You can easily do that with the metafor package in R.

After Table 2: No line numbers... so I am referring now to the name of subsections.

Subsection Key outcomes:

You said that the impact of age on the impact of the program differs between ages, would be great to have a quantification of this, a meta-analytic approach would be great if possible.  

Identified barriers: consider rephrasing the last sentence before Table 3.

Ethics: abused or died? what do you mean?

Reviewer 2 Report

Thank you for the opportunity to review an interesting review study.

The present review is the first to broadly map the concept, content and outcome of studies presenting family-based outdoor therapy for children and adolescents with mental health problems. The Authors identified seven studies targeting children and adolescents experiencing mental, emotional, developmental, behavioral, or social difficulties, and their families using different types of nature-based therapy interventions. Results from this study will inform researchers, practitioners, policymakers, and funders about experiences with family-based outdoor therapy programs.

I agree with the authors that due to the limited number of studies presenting family-based outdoor therapy programs for vulnerable children and adolescents, the variation in characteristics of participants, and methodological limitations of the presented studies, the evidence base for practice is scarce. 

The study is in line with the recommendations on page 16. From page 16 of the study not in a suitable format. I recommend editing.
